# Time-invariant working memory representations in the presence of code-morphing in the lateral prefrontal cortex

Aishwarya Parthasarathy [1], Cheng Tang [1], Roger Herikstad[2], Loong Fah Cheong[3], Shih-Cheng Yen [2,4,6]* & Camilo Libedinsky [1,2,5,6]*

Maintenance of working memory is thought to involve the activity of prefrontal neuronal populations with strong recurrent connections. However, it was recently shown that distractors evoke a morphing of the prefrontal population code, even when memories are maintained throughout the delay. How can a morphing code maintain time-invariant memory information? We hypothesized that dynamic prefrontal activity contains time-invariant memory information within a subspace of neural activity. Using an optimization algorithm, we found a low-dimensional subspace that contains time-invariant memory information. This information was reduced in trials where the animals made errors in the task, and was also found in periods of the trial not used to find the subspace. A bump attractor model replicated these properties, and provided predictions that were confirmed in the neural data. Our results suggest that the high-dimensional responses of prefrontal cortex contain subspaces where different types of information can be simultaneously encoded with minimal interference.

[1] Institute of Molecular and Cell Biology, A*STAR, Singapore, Singapore. [2] The N.1 Institute for Health, National University of Singapore (NUS), Singapore, Singapore. [3] Department of Electrical and Computer Engineering, NUS, Singapore, Singapore. [4] Innovation and Design Programme, Faculty of Engineering, NUS, Singapore, Singapore. [5] Department of Psychology, NUS, Singapore, Singapore. [6] These authors contributed equally: Shih-Cheng Yen, Camilo Libedinsky *email: shihcheng@nus.edu.sg; camilo@nus.edu.sg

Working memory (WM) is the ability to hold and manipulate information over a short time. It is a core component of complex cognitive functions, such as reasoning and language. Memory maintenance appears to involve the sustained activity of neurons in the lateral prefrontal cortex (LPFC)[1–3]; (but see[4–8]). Even though few neurons in the LPFC show persistent activity throughout the delay period[9–11], populations of LPFC neurons exhibit time-invariant (often referred to as stable in the literature) codes during the memory-maintenance period[12,13]. Distractors presented during WM maintenance disrupt code stability in the LPFC, despite behavioral evidence of WM stability[14,15]. If the LPFC plays a role in WM maintenance, then time-invariant memory information should be present in the morphing code.

We hypothesized that a LPFC response subspace retains time-invariant memory information in the presence of code-morphing. We used an optimization algorithm that minimized the subspace distance between the Delay 1 and Delay 2 responses using a cost function that included a penalty for information loss. Using this optimization, we found a subspace with a time-invariant memory code. The stability extended to the distractor presentation period, which was not used in the optimization, and the stability was absent in error trials. These results show that the LPFC retains behaviorally relevant time-invariant memory information despite exhibiting code-morphing.

Network models with strong recurrent connection between neurons with similar tuning have been shown to replicate several properties of LPFC activity, including code stability[16–19]. However, it is not known whether these models can exhibit code-morphing while retaining a subspace with time-invariant memory information. We found that a bump attractor model with memory and non-memory inputs was most effective in replicating the results we observed. These results suggest that non-memory inputs to the LPFC may be a critical component of code-morphing.

## Results

**Stable subspace**. Two adult monkeys were trained to perform a delayed saccade task with an intervening distractor (Fig. 1a). We recorded a total of 256 neurons from the LPFC and 137 neurons from the FEF while the animals performed the task. Cross-temporal decoding (Fig. 1b) and state-space analysis (Fig. 1c) showed that the distractor presentation led to code-morphing in the LPFC (quantified in Fig. 2d,e), as previously described[14] (see Supplementary Movie 1 for an illustration of the trajectories).

The presence of code-morphing presents an interesting decoding challenge: how can downstream regions read out time-invariant information from a morphing code? We hypothesized that there may be a low-dimensional subspace embedded in the LPFC population response that contained time-invariant memory information (Fig. 2a) that could be used by downstream cells. We used an optimization algorithm that minimized the distance between Delay 1 and Delay 2 responses when projected into a reference subspace, while simultaneously maintaining memory information (see Methods). The result of the optimization is shown in Fig. 2b, which shows overlapping Delay 1 and Delay 2 projections in the subspace. As hypothesized, classifiers trained in this subspace at one time point in either of the two delay periods were able to decode memory information equally well from other time points during both delay periods (Fig. 2c). We also found that the task-relevant target information was maintained in the subspace when compared to the full space (mean decoding performances in $LP_{11}$: full space—$69.75 \pm 0.08\%$ and subspace—$68.29 \pm 0.11\%$, $P \approx 0.95$; $LP_{22}$: full space—$70.95 \pm 0.09\%$, and subspace—$70.41 \pm 0.19\%$, $P \approx 0.92$ (permutation test)). In order to quantify the stability of the code in the subspace, we calculated the difference in decoding performance between decoders trained and tested in Delay 1 ($LP_{11}$), and decoders trained in Delay 1 and tested in Delay 2 ($LP_{12}$). Similarly, we compared the performance difference for decoders trained and tested in Delay 2 ($LP_{22}$), and decoders trained in Delay 2 and tested in Delay 1 ($LP_{21}$). In this analysis, values above zero implied code-morphing, while values around zero implied a time-invariant code. For decoders built using the full space, we found significant differences, consistent with code-morphing (Fig. 2d, light gray, $P < 0.001$ for $LP_{11} - LP_{12}$ and $LP_{22} - LP_{21}$). On the other hand, for decoders built using the subspace, we found no difference, consistent with a time-invariant code (Fig. 2d, dark gray, $P \approx 0.17$ (permutation test) for $LP_{11} - LP_{12}$ and $P \approx 0.08$ (permutation test) for $LP_{22} - LP_{21}$).

We also quantified the stability of the code using state space analysis by calculating the mean shift in cluster centers between Delay 1 and Delay 2 (inter-delay shift), compared to the mean intra-delay shift in Delays 1 and 2 (Fig. 2e). In this analysis, inter-delay shifts that were larger than the intra-delay shifts implied code-morphing, while similar shifts implied a time-invariant code. In the full space (Fig. 2e, light gray), we found significantly larger inter-delay shifts when compared to intra-delay shifts ($P < 0.001$ (permutation test), $g = 7.40$), consistent with code-morphing. On the other hand, in the subspace (Fig. 2e, dark gray), no such differences were found ($P \approx 0.84$ (permutation test)), consistent with a time-invariant code.

In order to determine the dimensionality of the full space and the subspace, we computed the cumulative percent variance explained by different numbers of PCA components (Fig. 2f). In the full space, the top six components were required to explain at least 95% of the variance, while in the subspace, only the first component was required to explain at least 95% of the variance. This supported the view that the subspace was clearly different from the full space, and existed in a lower dimensional space within the full state-space.

The optimization algorithm used in this study to identify the time-invariant subspace used activity from the last 500 ms of Delays 1 and 2. We selected these periods because both exhibited internal stability. To our surprise, the subspace stability extended to periods that were not used in the optimization, namely distractor period ($P \approx 0.27$ (permutation test) for decoding performance difference; $P \approx 0.96$ (permutation test) cluster shifts) and the first 500 ms of Delay 2 ($P \approx 0.33$ (permutation test) for decoding performance difference; $P \approx 0.19$ (permutation test) for cluster shifts; Fig. 2c, e). We also showed that the presence of stable subspace when the optimization algorithm was applied to data from individual monkeys (Supplementary Fig. 1). A subspace built using four out of seven target locations also contained time-invariant information about the 3 locations not included in the optimization, supporting the notion that the identified subspace generalizes to all spatial locations (Supplementary Fig. 2). The stability of the subspace, however, did not extend to the target presentation period ($P < 0.001$ (permutation test) for decoding performance difference; $P < 0.001$ (permutation test), $g = 6.37$ for cluster shifts), nor the first 500 ms of Delay 1 ($P < 0.001$ (permutation test) for decoding performance difference; $P < 0.001$ (permutation test), $g = 3.94$ for cluster shifts; Fig. 2c). This observation may reflect a shortcoming of our method in identifying the true subspace that was used from the moment of stimulus presentation until the response. Alternatively, it may reflect a real specialization of the LPFC in encoding memories ~800 ms after target presentation[13,15,18], which may be the time when activity in the subspace we identified started encoding memory information (Fig. 2c). This interpretation is consistent with our inability to find a time-invariant subspace

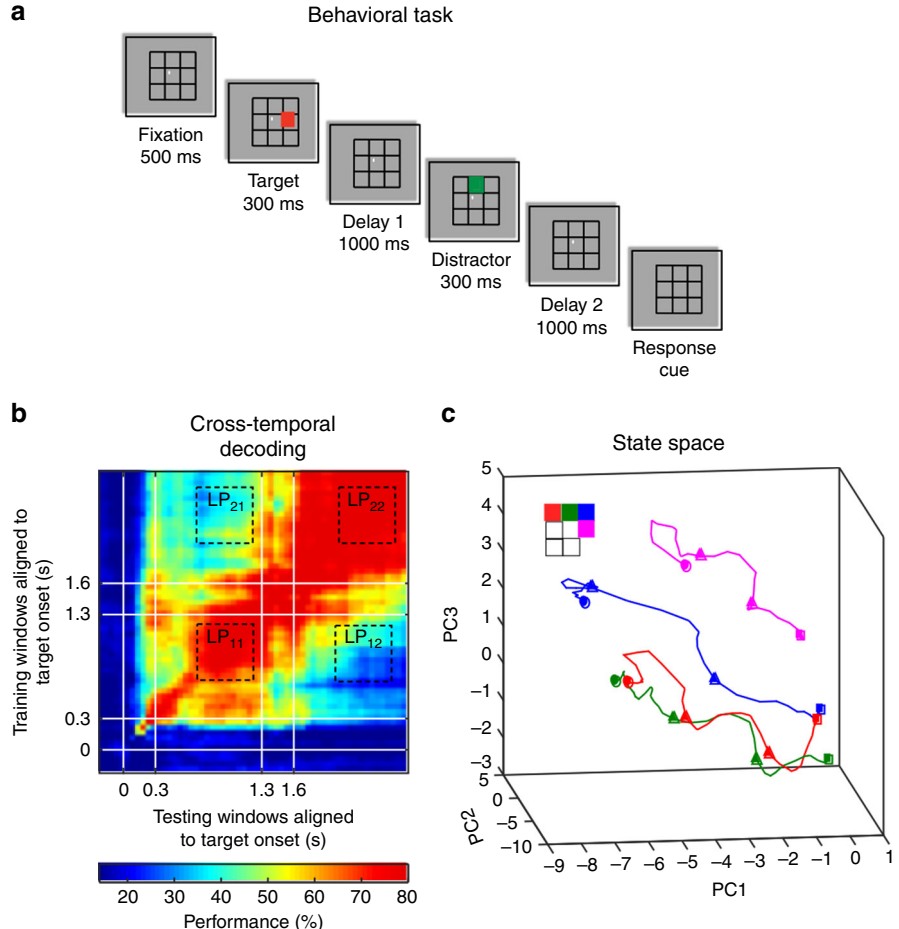

**Fig. 1** Experimental design and code-morphing. **a** Behavioral task: each trial began when the animal fixated on a fixation spot in the center of the screen. The animal was required to maintain fixation throughout the trial until the fixation spot disappeared. A target (red square) was presented for 300 ms followed by a 1000-ms delay period (Delay 1). A distractor (green square) was then presented for 300 ms in a random location that was different from the target location, and was followed by a second delay of 1000 ms (Delay 2). After Delay 2, the fixation spot disappeared, which was the Go cue for the animal to report, using an eye movement, the location of the target. **b** Heat map showing the cross-temporal population-decoding performance in the LPFC. White lines indicate target presentation (0–0.3 s) and distractor presentation (1.3–1.6 s). **c** Responses of the LPFC population when projected onto the first three principal components (PC) of the combined Delay 1 and Delay 2 response space. The responses for different target locations are color-coded using the color scheme shown in the top left. The trajectories illustrate the evolution of the responses from 500 ms before the end of Delay 1 (square), distractor onset (first triangle), distractor offset (second triangle), and the end of Delay 2 (circle). The trajectories of only four of the seven target locations are shown here for clarity

when performing the optimization using target period and Delay 2 period activities (Supplementary Fig. 3).

Code-morphing depended on the activity of neurons with mixed selectivity[20], which in our task were defined as those that exhibited simultaneous selectivity to multiple task parameters, such as memory location and task epoch[14]. Of particular relevance for code-morphing appeared to be those neurons with non-linear mixed selectivity (NMS), which exhibited an interaction between the selectivity to target location and task epoch. In other words, NMS neurons had different selectivity before and after the distractor[14]. A fraction of the neurons in LPFC were classically selective, meaning that they were only selective to memory location or task epoch, but not both, and did not change their selectivity after the distractor. One possible interpretation of the subspace could be that it was built from the activity of classically selective neurons, which by definition would have a time-invariant code. While this would be a simple explanation for the existence of the subspace, it was unlikely, since classically selective cells contained comparatively little information about the target location, likely due to their poor selectivity

(Supplementary Fig. 4a–c). Furthermore, individual neuron contributions to the subspace were similarly distributed across NMS, classically selective, and linear mixed selective cells (Supplementary Fig. 4d). A subspace could also be identified in a population exclusively composed of NMS neurons, highlighting that the classically selective neurons were not essential to the formation and maintenance of time-invariant memory information (Supplementary Fig. 5).

**Parallel trajectories in the full space.** The results discussed so far showed that time-invariant memory information could be extracted from a subspace of LPFC neurons exhibiting code morphing. In order to understand whether this finding reflected a coding property of LPFC neurons, or whether any random set of state-space trajectories contain a subspace with time-invariant memory information, we carried out the same analysis on data where the memory locations in Delay 2 were shuffled with respect to Delay 1 (Fig. 3a, b). Using the shuffled data, we were not able to find a subspace that provided a time-invariant memory

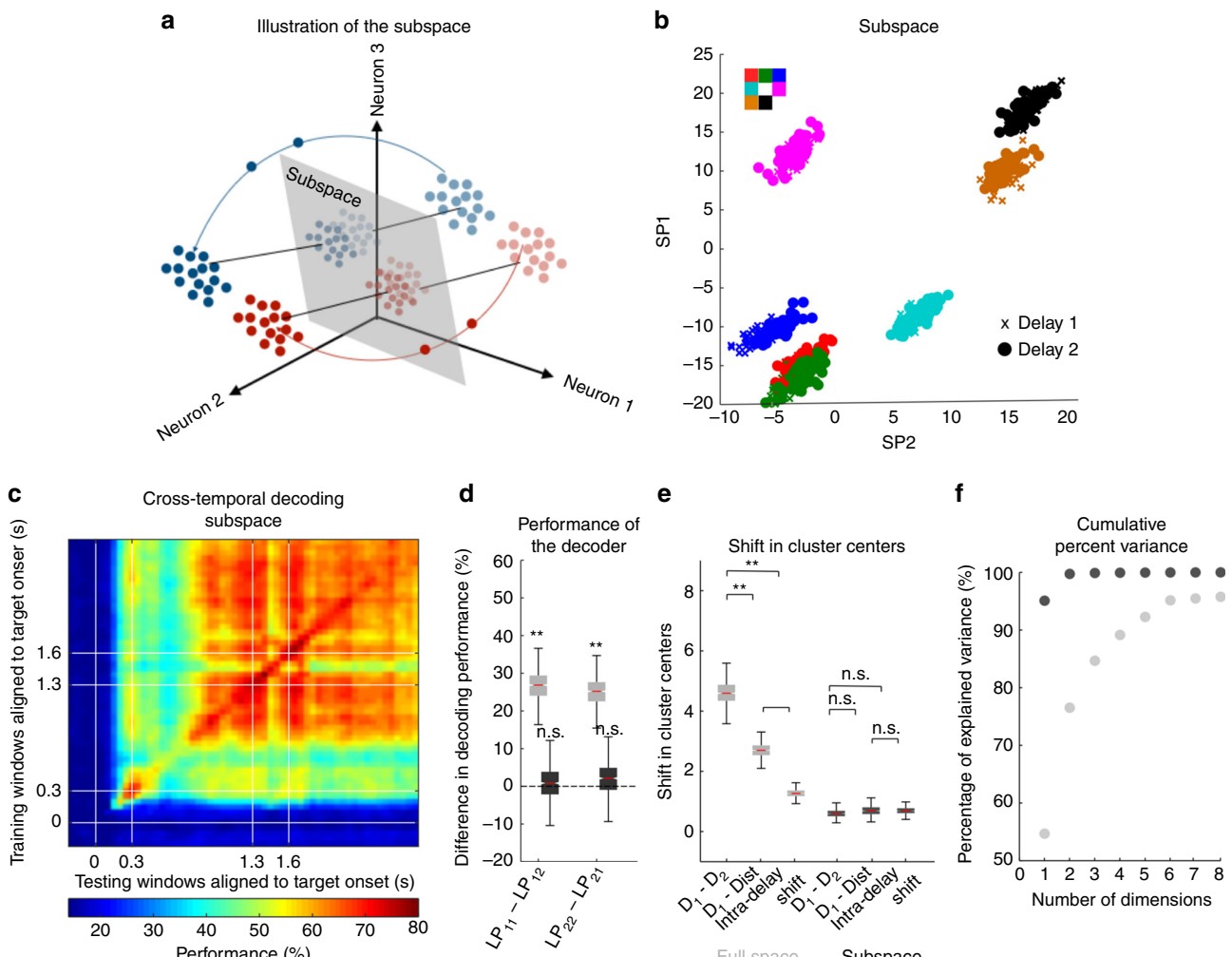

**Fig. 2** Identification of a subspace. **a** Illustration of the subspace where Delay 1 and Delay 2 activity exists as one persistent code. **b** Delay 1 (plotted using crosses) and Delay 2 (plotted using squares) responses after projection into the top 2 dimensions of the subspace. Points for different target locations are color-coded according to the color scheme shown in the top left. **c** Heat map showing the cross-temporal population-decoding performance after the population responses were projected onto the subspace. White lines indicate target presentation (0–0.3 s) and distractor presentation (1.3–1.6 s). **d** The difference in decoding performance when a decoder that was trained on Delay 1 responses was tested on Delay 1 ($LP_{11}$) or Delay 2 ($LP_{12}$) responses are shown in the gray box-plot labelled $LP_{11}$–$LP_{12}$, and vice versa (gray box-plot labelled $LP_{22}$–$LP_{21}$). The equivalent performance differences in the subspace are shown in the black box-plots. **e** The shift in cluster centers from Delay 1 to Delay 2 (labeled $D_1$–$D_2$) averaged across target locations, from Delay 1 to the distractor presentation period ($D_1$ - Dist), and the intra-delay shifts in both Delays 1 and 2 (intra-delay) are shown in the full space (gray box-plot), and in the subspace (black box-plot). **f** The cumulative explained variance is plotted as a function of the number of PCs for the full space (plotted in gray) and the subspace (plotted in black). The explained variance denoted the dimensionality of the projections in the full space and the subspace (not to be confused with the 58 dimensions that explained 90% of the variance in normalized z-scored data). This was performed to show that the dimensionality of the subspace was lower than that of the full space. The bounds in the boxplots are defined by the 25th and 75th percentile of the distribution. The red line represents the median and the whiskers represent the 2.5th and 97.5th percentile of the distribution

readout (Fig. 3c, d). The optimization procedure yielded a subspace with low information content (Fig. 3d), and which still displayed code-morphing (Fig. 3e, f; $LP_{11} - LP_{12}$: $P < 0.001$ (permutation test); $LP_{22} - LP_{21}$: $P < 0.001$ (permutation test); shift in cluster centers: true data versus intra-delay shift: $P \approx 0.96$ (permutation test); shuffled data versus intra-delay shift: $P < 0.04$ (permutation test), $g = 3.58$). These results imply that the existence of a subspace with time-invariant memory information reflects a non-trivial organizational property of LPFC activity. In order to characterize this organizational property we employed a dynamical systems approach, by analyzing the trajectory dynamics of population activity in state space[21]. For the subspaces created using true and shuffled data, we calculated the average trajectory directions, which is the magnitude of the vector

obtained by averaging the trajectory vectors from Delay 1 to Delay 2 for each target location. In this analysis, low values of trajectory direction would be expected from trajectories that move in non-parallel ways. We found that trajectories in the subspace built using true data moved in significantly more parallel trajectories than in the subspace built using shuffled data (Fig. 3g, $P < 0.001$ (permutation test), $g = 1.74$). This suggested that the parallel movement of trajectories could be an important response property of LPFC activity that facilitated the existence of a subspace with time-invariant memory information.

**Stable subspace during error trials.** Although we were able to identify a subspace in which the target information could be stably decoded throughout Delays 1 and 2 in spite of

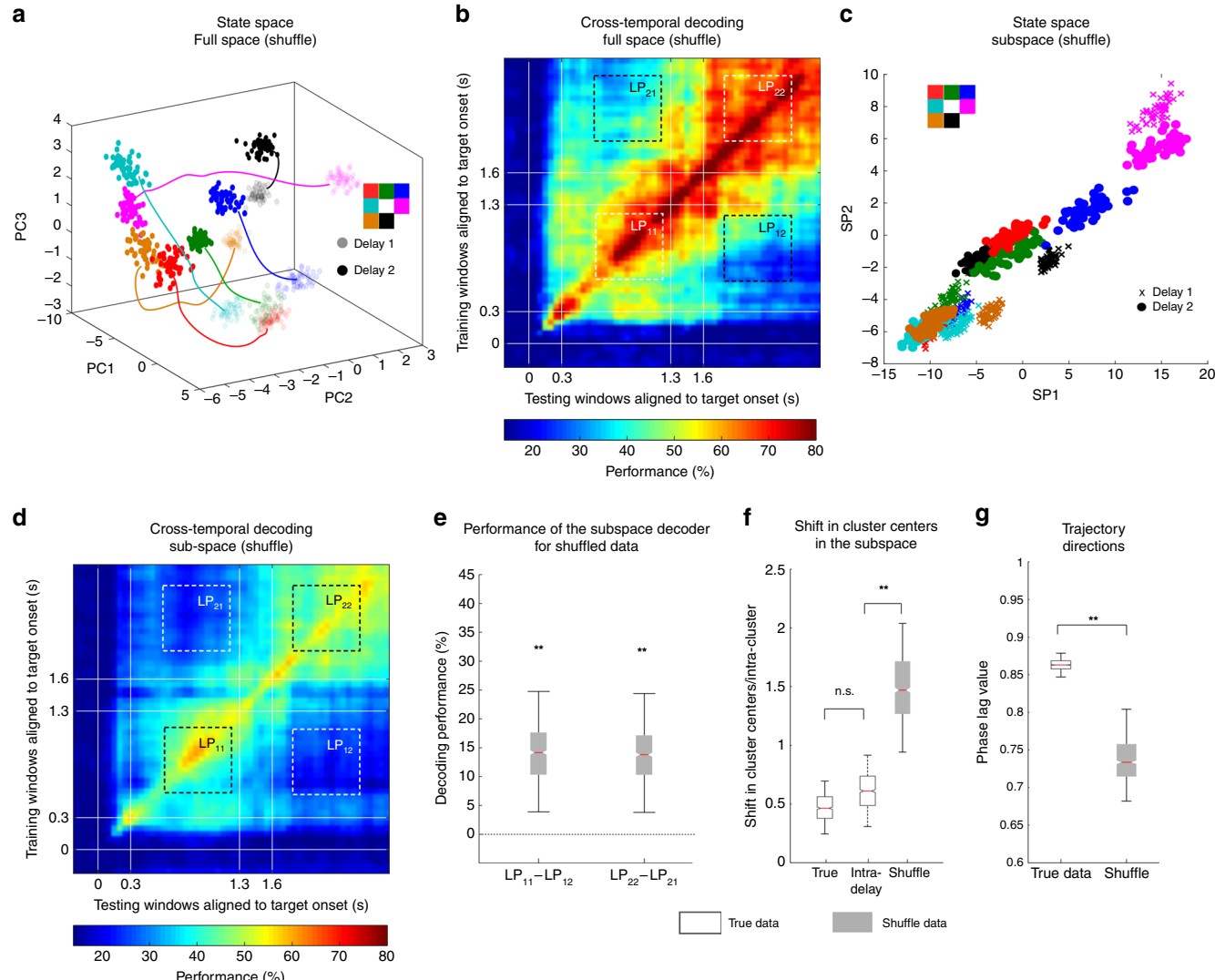

**Fig. 3** Comparison of the subspace to a shuffled subspace. **a** State space showing the target locations in Delay 2 (dark colored dots) were shuffled relative to those in Delay 1 (light colored dots) before the optimization was performed in the full space. **b** Heat map showing the cross-temporal population-decoding performance in the full space with the target location in Delay 2 shuffled relative to those in Delay 1. **c** Delay 1 (plotted using crosses) and shuffled Delay 2 (plotted using open circles) responses after projection into the top 2 dimensions of the subspace. Points for different target locations are color-coded according to the color scheme shown in the top left. **d** Heat map showing the cross-temporal population-decoding performance in the subspace with the target locations in Delay 2 shuffled relative to those in Delay 1. **e** The difference in decoding performance when a decoder that was trained on the Delay 1 responses shown in **c** was tested on Delay 1 ($LP_{11}$) or Delay 2 ($LP_{12}$) responses are shown in the gray box-plot labelled $LP_{11}$ - $LP_{12}$, and vice versa (labelled $LP_{22}$ - $LP_{21}$). **f** The shift in cluster centers from Delay 1 to Delay 2 in the subspace, normalized to the average intra-delay distance, is shown for the true data (black box-plot) and the shuffled data (gray box-plot). The boxplot in the middle illustrates the intra-delay shifts for comparison. **g** The magnitude of the vector obtained by averaging the trajectory vectors from Delay 1 to Delay 2 for each target location is shown for the true data (black box-plot) and the shuffled data (gray box-plot). The bounds in the boxplots are defined by the 25th and 75th percentile of the distribution. The red line represents the median and the whiskers represent the 2.5th and 97.5th percentile of the distribution

code-morphing, it was not clear whether this subspace was related to the behavioral performance of the animals in the task. We investigated this question by comparing the responses of correct and incorrect trials in the subspace. The cross-temporal decoding performance for error trials showed that less information can be decoded within the subspace (Fig. 4a). Compared to the performance for correct trials shown in Fig. 2c, the error trials clearly exhibited much lower performance. We found the performance difference between correct and incorrect trials in the subspace was not significantly different from that in the full space (Fig. 4b) for both Delays 1 ($P ≈ 0.61$ (permutation test)) and 2 ($P ≈ 0.14$ (permutation test)). By analyzing the shift in cluster centers in the subspace, we found that error trials began to deviate from correct trials in Delay 1 (although the deviation did not reach statistical significance, $P ≈ 0.08$ (permutation test), and become significantly different from correct trials in Delay 2 (Fig. 4c, $P < 0.001$ (permutation test), $g = 3.16$). These results show that behavioral errors were associated with decreases of information within the subspace.

While some errors may be driven by failures of working memory, as reflected in the information decay within the subspace, other factors may also lead to errors, such as failures in motor preparation or alertness. Presumably, these other factors would be reflected in information decay in spaces

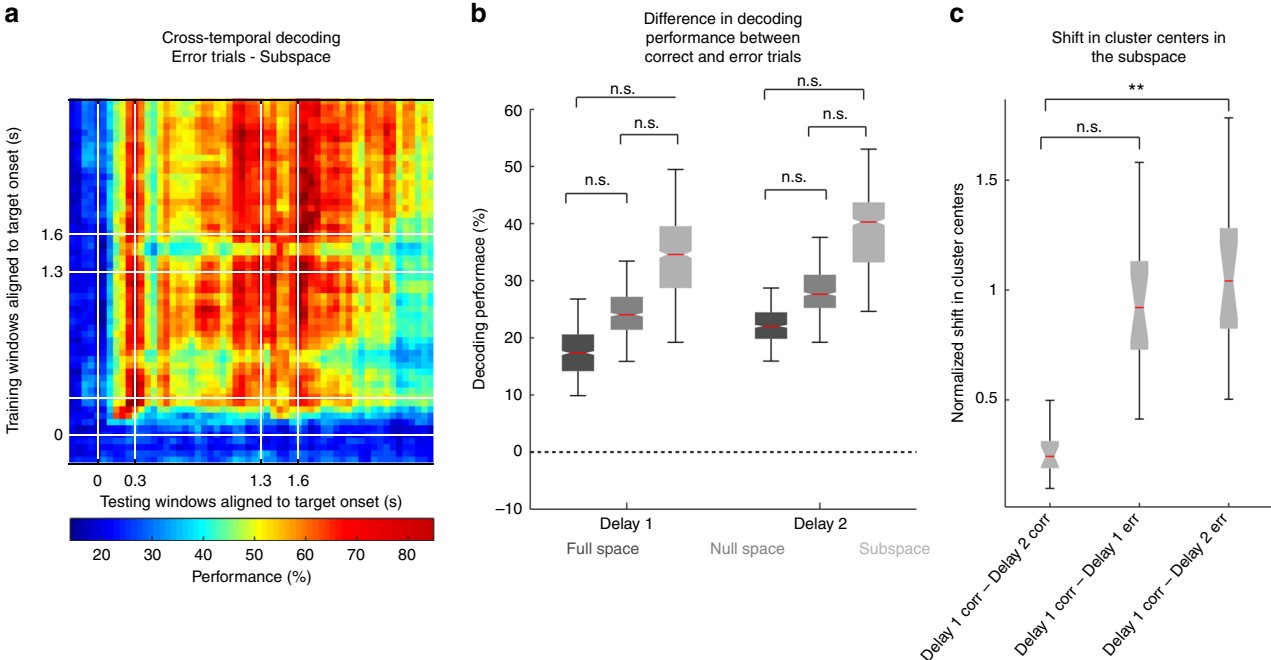

**Fig. 4** Subspace decoding in error trials. **a** Heat map showing the cross-temporal population-decoding performance in the subspace for error trials (only four target locations with high error rates are included). **b** The differences in decoding performance between correct and error trials are shown in the full space (black box-plot), subspace (gray box-plot), and the null space (dark gray box-plot) for both Delays 1 and 2. **c** The normalized shift in cluster centers in the subspace is shown for correct trials between Delay 1 to Delay 2 (black box-plot), between correct and error trials in Delay 1 (dark gray box-plot), and between correct trials in Delay 1 and error trials in Delay 2 (gray box-plot). The bounds in the boxplots are defined by the 25th and 75th percentile of the distribution. The red line represents the median and the whiskers represent the 2.5th and 97.5th percentile of the distribution

orthogonal to the memory subspace (i.e. the null space). In order to test this hypothesis, we compared the changes in decoding performance in the subspace and the null space. As hypothesized, error trials also showed decreased target information in the null space (Fig. 4b). While this decrease in target information in error trials in the null space was smaller than that seen within the memory subspace, this difference was not significant (Fig. 4b). Thus, while these results were consistent with the interpretation that reduced information within the subspace can lead to behavioral errors, there was insufficient evidence to suggest that this subspace was directly linked to the behavior of the animal.

**Bump attractor model best describes the stable subspace**. A recent study showed that time-invariant memory information could be decoded from LPFC activity despite the complex and heterogeneous temporal dynamics of single-neuron activity[18]. They applied PCA to the time-averaged delay activity across stimulus conditions to find a mnemonic subspace in a working memory task that did not contain intervening distractors[18]. When we applied the same method to our data, we also found a time-invariant subspace prior to distractor presentation. However, after distractor presentation, the code also morphed in this subspace (Supplementary Fig. 6). Another method we considered to identify a time-invariant subspace involved using LDA. The subspace identified using LDA could be used to read out time-invariant memory information in both delays. However, within the LDA subspace, there were significant shifts in clusters between Delay 1 and Delay 2 projections (Supplementary Fig. 7), suggesting that the apparent stability observed with decoding would break down if the task required discrimination using finer-grained memories.

Artificial neural network models (ANNs) have been shown to replicate several properties observed in the LPFC during working memory tasks[16–18]. However, the observation of the existence of a time-invariant subspace in the presence of code-morphing imposes new constraints on these models.

Here, we tested two different models to attempt to replicate the following list of observations: (1) time-invariant memory code within Delay 1 and within Delay 2, (2) code-morphing after distractor presentation, (3) existence of non-linear mixed selective neurons, and (4) existence of a time-invariant subspace. Since the parameter space for recurrent neural networks was very large, and as Murray et al.[18] had reported that certain trained recurrent neural networks (RNN) with chaotic activity did not lead to stable subspace representations, we chose to compare two other types of ANNs that were likely to display these properties: a bump attractor model[17], and a linear subspace model[18]. Initially, we tested the models assuming that the inputs received during distractor presentation were similar or weaker than those used during target presentation (to emulate the low behavioral relevance of the distractor in our task). Under these input parameters, neither of the models were able to replicate all the properties listed above (results are shown for the bump attractor model in Supplementary Fig. 8). We hypothesized that additional non-memory inputs during the distractor presentation may be required to replicate these properties. These additional inputs could be interpreted as ascending modulatory inputs, thalamic inputs, or inputs that encoded additional information, such as movement preparation or reward expectation. Using this additional input, we found that only the bump attractor model allowed us to replicate all these properties (Fig. 5), while the linear subspace model failed to replicate the predominance of neurons with non-linear mixed selectivity (Supplementary Fig. 9). The connections in the bump

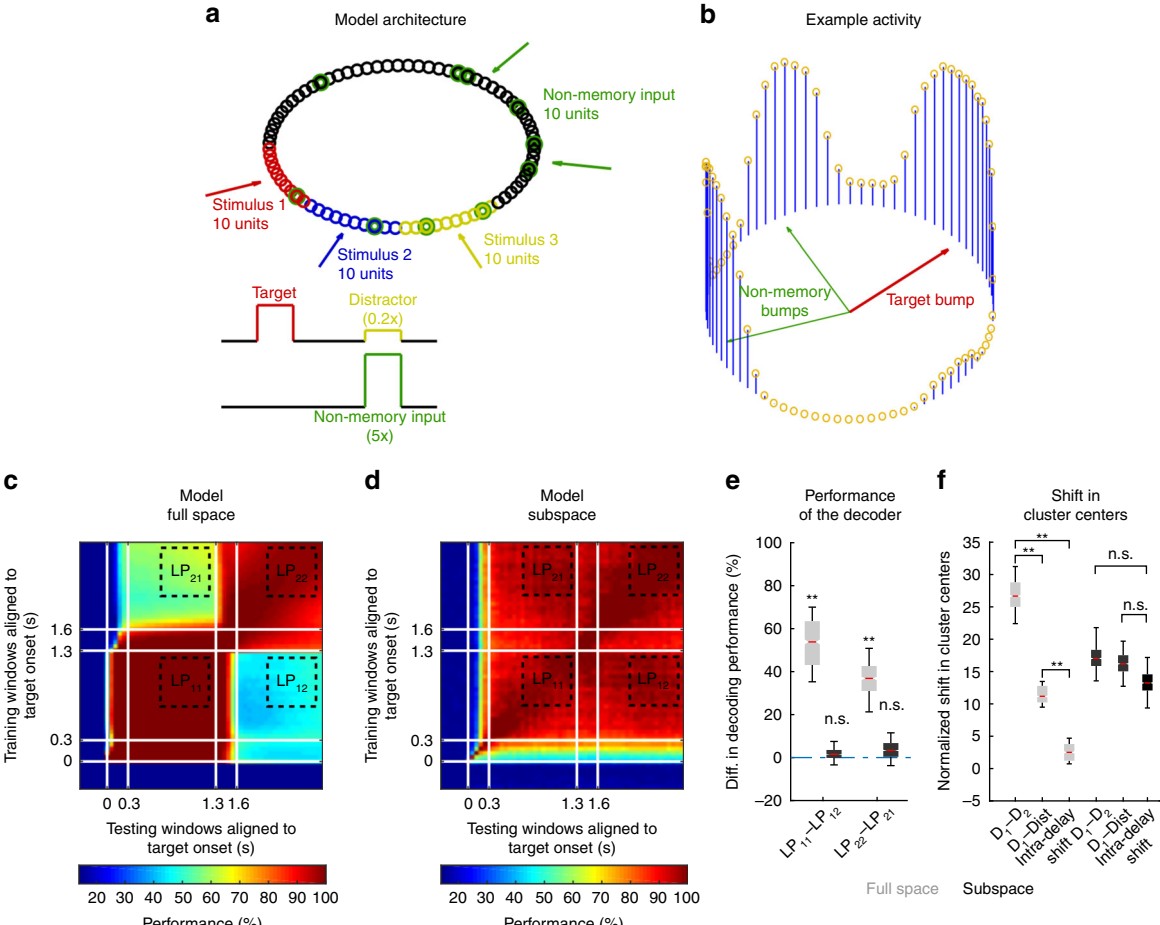

**Fig. 5** Model of code-morphing in prefrontal cortex. **a** Bump attractor model (adapted from Wimmer et al.[17]) with eight location inputs representing the eight location stimuli for target or distractor, and a non-memory input that was active during the distractor presentation period. The recurrent layer contained 80 units, and each location input projected onto 10 adjacent units that were non-overlapping for different locations (red, blue, and yellow units). The non-memory input projected onto 10 random units in the population (green units), which overlapped with different location units. **b** An example of the activity found in the recurrent layer in Delay 2 after the distractor was presented. **c** Heat map showing the cross-temporal population-decoding performance of the model in the full space. **d** Heat map showing the cross-temporal population-decoding result of the model after projecting into the subspace. **e** The differences in decoding performance ($LP_{11} - LP_{12}$, $LP_{22} - LP_{21}$) in the model in the full space are shown in gray box-plots. The equivalent performance differences in the subspace are shown in the black box-plots. **f** The normalized shift in cluster centers averaged across target locations ($D_1$–$D_2$, $D_1$—Dist, and intra-delay) in the model are shown in the full space (gray box-plot), and in the subspace (black box-plot). The bounds in the boxplots are defined by the 25th and 75th percentile of the distribution. The red line represents the median and the whiskers represent the 2.5th and 97.5th percentile of the distribution

attractor model consisted of strong excitation to neighboring units, and weak inhibition to units further away (see Methods). This architecture ensured a time-invariant code during the first delay, but did not exhibit code-morphing after the distractor presentation. However, with the addition of the non-memory inputs, code-morphing occurred due to the addition of new bumps of activity after the distractor presentation, and the locations of these new bumps were a function of the non-memory inputs (Fig. 5b). The optimization algorithm successfully identified a time-invariant subspace from the responses in this model (Fig. 5c, d), and a state space analysis verified that the model's responses exhibited dynamics consistent with the neural data in the full space and the subspace (Fig. 5e, f). A range of non-memory input parameters led to code-morphing in the model, suggesting that there was some degree of flexibility in the non-memory input parameters (Supplementary Fig. 10). The model also provided a couple of testable predictions: first, that the initial bump would be maintained in Delay 2; and second, that the response fields in Delay 2 would be larger than those in

Delay 1. Both predictions were corroborated in the neural data (Supplementary Fig. 11a, b).

Although the exact nature of the non-memory input was not clear, the model provided predictions to differentiate between different possibilities using a task with two consecutive distractors, separated by 1 s each (rather than the one distractor we used in the experiment). If the non-memory input corresponded to ascending modulatory or thalamic inputs that were triggered after every distractor, then code-morphing would occur after the first distractor, but not the second since the same non-memory inputs would be activated (Supplementary Fig. 11c). On the other hand, if the non-memory input corresponded to a movement preparation or reward expectation signal triggered by the stimulus that best predicted the timing of movement or reward onset, then code-morphing would occur after the second distractor (which was closest to the movement and reward onset), but not after the first (Supplementary Fig. 11d). Furthermore, target information and non-memory information were encoded in a mixed manner in the same population in our model, which

contrasted with a recent modelling study that showed different types of information were spontaneously encoded by different clusters of units[19]. These would be interesting predictions to test in future experiments.

## Discussion

Here we demonstrate that time-invariant memory information can be read out from a population of neurons that morphs its code after a distractor is presented. This readout was enabled by a low-dimensional subspace, which was identified using an optimization algorithm that minimized the distance between projections of Delay 1 and Delay 2 activities onto state-space, while minimizing information loss. We found that an important property of neural dynamics that allowed for the existence of this subspace was the parallel movement of trajectories from Delay 1 to Delay 2 for different memory locations. Information in this subspace appears to be behaviorally relevant, since the stability breaks down in error trials. Finally, a bump attractor model replicated code-morphing and time-invariant subspace, and revealed that code-morphing required a non-memory input during the distractor presentation period. Overall, our results show that dynamic activity in LPFC, possibly driven by non-memory inputs during distractor presentation, can be read out in a time-invariant manner to perform a task that requires time-invariant working memory information.

The LPFC has a high number of neurons with mixed-selective responses, which dramatically increase the dimensionality of representations[20], and may support reliable information transmission[22]. The existence of multiple subspaces within a single population of neurons may be an efficient means to use the high-dimensional activity space of brain regions[21]. This view is consistent with our finding that a low-dimensional subspace within LPFC can encode time-invariant memory information despite the presence of neural dynamics caused by the addition of new information. We used the term non-memory input because there was no additional memory in our task, but the new information could be the addition of a second memory item in a different task. Additional dynamics, which may reflect activity in orthogonal spaces, would allow simultaneous encoding of additional information without interfering with existing memory information. However, orthogonality of subspaces is not a necessity, since different types of information may interfere with each other at the cognitive and neural levels. For example, attention or movement preparation may bias or interfere with working memory information[23–25], which would suggest their encoding in subspaces, which are not orthogonal to the memory subspace.

We have shown here that, in principle, a region downstream from the LPFC could readout time-invariant memory information using the subspace. However, we have not shown that any regions are actually using this subspace to read out time-invariant memory information. Addressing this question is particularly challenging, since even if a downstream region reads out this information, it may be immediately converted to a new task-relevant type of information, such as direction of eye movement. In order to assess whether downstream regions indeed use the memory subspace of LPFC we could assess how trial-to-trial fluctuations of population responses relate the LPFC subspace with fluctuations in downstream areas[26]. However, a more direct test would involve the manipulation of LPFC activity, either within or outside the subspace, while measuring changes in activity in downstream regions. Unfortunately, these experiments would not be feasible with sti-mulation technologies available today. One approach to address this question is to generate artificial neural network models that replicate the response properties of LPFC and downstream regions, and to generate predictions of the effect of specific

manipulations, that could be tested in biological data. Here we tested two different types of artificial neural network models to try to replicate the existence of a time-invariant memory subspace in the presence of code-morphing. We found that the bump attractor model could replicate these features, while a subspace model failed to replicate these behaviors. The model provided predictions that could be corroborated on the data, and other predictions that future studies could be able to test. The approach presented here could help reconcile the apparent incompatibility between LPFC activity changes after behaviorally relevant distractors and attractor network models[15]. These observations support this model as a useful abstraction of the LPFC function.

## Methods

**Subjects and surgical procedures**. We used two male adult macaques (*Macaca fascicularis*), Animal A (age 4) and Animal B (age 6), in the experiments. All animal procedures were approved by, and conducted in compliance with the standards of the Agri-Food and Veterinary Authority of Singapore and the Singapore Health Services Institutional Animal Care and Use Committee (SingHealth IACUC #2012/SHS/757). The procedures also conformed to the recommendations described in Guidelines for the Care and Use of Mammals in Neuroscience and Behavioral Research (National Academies Press, 2003). Each animal was implanted first with a titanium head-post (Crist Instruments, MD, USA) before arrays of intracortical microelectrodes (MicroProbes, MD, USA) were implanted in multiple regions of the left frontal cortex (Fig. 1c). In Animal A, we implanted 6 arrays of 16 electrodes and 1 array of 32 electrodes in the LPFC, and two arrays of 32 electrodes in the FEF, for a total of 192 electrodes. In Animal B, we implanted one array of 16 electrodes and two arrays of 32 electrodes in the LPFC, and two arrays of 16 electrodes in the FEF, for a total of 112 electrodes. The arrays consisted of platinum-iridium wires with either 200 or 400 μm separation, 1–5.5 mm of length, 0.5 MΩ of impedance, and arranged in 4 × 4 or 8 × 4 grids. Surgical procedures followed the following steps. Twenty-four hours prior to the surgery, the animals received a dose of Dexamethasone to control inflammation during and after the surgery. They also received antibiotics (amoxicillin 7–15 mg/kg and Enrofloxacin 5 mg/kg) for 8 days, starting 24 h before the surgery. During surgery, the scalp was incised, and the muscles retracted to expose the skull. A craniotomy was performed (~2 × 2 cm). The dura mater was cut and removed from the craniotomy site. Arrays of electrodes were slowly lowered into the brain using a stereotaxic manipulator. Once all the arrays were secured in place, the arrays' connectors were secured on top of the skull using bone cement. A head-holder was also secured using bone cement. The piece of bone removed during the craniotomy was repositioned to its original location and secured in place using metal plates. The skin was sutured on top of the craniotomy site, and stitched in place, avoiding any tension to ensure good healing of the wound. All surgeries were conducted using aseptic techniques under general anesthesia (isofluorane 1–1.5% for maintenance). The depth of anesthesia was assessed by monitoring the heart rate and movement of the animal, and the level of anesthesia was adjusted as necessary. Analgesics were provided during post-surgical recovery, including a Fentanyl patch (12.5 mg/2.5 kg 24 h prior to surgery, and removed 48 h after surgery), and Meloxicam (0.2–0.3 mg/kg after the removal of the Fentanyl patch). Animals were not euthanized at the end of the study.

**Recording techniques**. Neural signals were initially acquired using a 128-channel and a 256-channel Plexon OmniPlex system (Plexon Inc., TX, USA) with a sampling rate of 40 kHz. The wide-band signals were band-pass filtered between 300 and 3000 Hz. Following that, spikes were detected using an automated Hidden Markov Model based algorithm for each channel[27]. The eye positions were obtained using an infrared-based eye-tracking device from SR Research Ltd. (Eyelink 1000 Plus). The behavioral task was designed on a standalone PC (stimulus PC) using the Psychophysics Toolbox in MATLAB (Mathworks, MA, USA). In order to align the neural and behavioral activity (trial epochs and eye data) for data analysis, we generated strobe words denoting trial epochs and performance (rewarded or failure) during the trial. These strobe words were generated on the stimulus PC, and were sent to the Plexon and Eyelink computers using the parallel port.

**Behavioral task**. Each trial started with a mandatory period (500 ms) where the animal fixated on a white circle at the center of the screen. While continuing to fixate, the animal was presented with a target (a red square) for 300 ms at any one of eight locations in a 3 × 3 grid. The center square of the 3 × 3 grid contained the fixation spot and was not used. The presentation of the target was followed by a delay of 1000 ms, during which the animal was expected to maintain fixation on the white circle at the center. At the end of this delay, a distractor (a green square) was presented for 300 ms at any one of the seven locations (other than where the target was presented). This was again followed by a delay of 1000 ms. The animal was then given a cue (the disappearance of the fixation spot) at the end of the second delay to make a saccade towards the target location that was presented earlier in the trial. Saccades to the target

location within a latency of 150 ms and continued fixation at the saccade location for 200 ms was considered a correct trial. An illustration of the task is shown in Fig. 1a. One of the animals was presented with only seven of the eight target locations because of a behavior bias in the animal.

**Firing rate normalization**. The firing rate of each neuron (averaged across trials with 100 ms windows with 50 ms of overlap) was converted to a z-score by normalizing to the mean and standard deviation of the instantaneous firing rates from 300 ms before target onset to target onset. These z-scores were then used for the state-space and subspace analyses. Our database initially consisted of 256 LPFC neurons, but we excluded 12 neurons as they exhibited responses that were very similar to other neurons, suggesting that they were the result of over-clustering during spike sorting. All the analysis on experimental data was performed on a pseudo-population built from data collected from two monkeys. Due to an unequal distribution of error trials across different locations, we only analyzed the error trials from four target locations. We only included locations in the analysis where there were at least six error trials in every session.

**Subspace identification**. We used the following optimization equation to identify the subspace:

$$\underset{U}{argmin}\left( \left\| U(D_1 - D_2) \right\|_F - \frac{\beta}{2}\left( \left\| UD_1^{av} \right\|_F + \left\| UD_2^{av} \right\|_F \right) \right) \quad (1)$$

where the $\|\cdot\|_F$ notation referred to the Frobenius Norm. We postulated that there was a matrix transformation, $U$, that would be able to transform both the Delay 1 ($D_1$) and Delay 2 ($D_2$) responses to a subspace where the distance between the corresponding target responses in the two delays were minimized. Additionally, to make sure that the information about the target location was preserved in this subspace, we added two terms $UD_1^{av}$ and $UD_2^{av}$, where $D_1^{av}$ and $D_2^{av}$ referred to the mean subtracted population activity. The multiplier $\beta$ was introduced to weigh maximization of information differentially from minimization of the distance between $D_1$ and $D_2$. We chose $\beta$ to be 0.1 as it repeatedly yielded subspaces with stability, as well as information about target locations. In practice, we first reduced the normalized firing rates from 800 ms to 2500 ms after target onset (i.e. 500 ms before the end of $D_1$ to the end of $D_2$) of the 244 LPFC neurons to a smaller number of Principal Components Analysis (PCA) components that accounted for 90% of the variance, which turned out to be 58. We subsequently used these 58 dimensions to be the full space in the remainder of this paper. We then took 50 responses for each of the seven target locations to create $D_1$ and $D_2$ matrices that were $58 \times 350$ in size, where each of the columns was the averaged z-score for the 244 cells over the last 500 ms of $D_1$ or $D_2$ for one target location projected onto the 58 components. $U$ was then a $u_{dim} \times 58$ matrix (where $u_{dim}$ was a parameter that determined the number of dimensions in the subspace), and was initialized with random values.

The optimization was performed using the *fmincon* function in Matlab to minimize the cost function shown above using the sequential quadratic programming (sqp) algorithm. There were no additional constraints imposed on this optimization. The results shown in Figs. 2–4 contain results from an optimized eight-dimensional subspace (i.e. $u_{dim} = 8$). We also performed this optimization using a 9- and 10-dimensional subspace, and found that the subspace was not qualitatively different for 8, 9, or 10 dimensions. Thus, we chose the conservative option of 8 dimensions to verify the stability of the subspace.

Figure 2b shows the projection $UD_1$ and $UD_2$ in the first 2 dimensions of the subspace $U$. We named these dimensions $SP_1$ and $SP_2$. Further, we concatenated $U$ x $D_1$ and $U$ x $D_2$ together to create a matrix that was $8 \times 700$, and then performed PCA again to obtain the cumulative contribution of each dimension to variance (dimensionality) in this subspace $U$ and in the full space, as shown in Fig. 2f. Note that we only show the contribution of the first eight dimensions of the full space in Fig. 2f.

In order to compute the contribution of each neurons, we took the $8 \times 58$ $U$ matrix, and multiplied it by the $58 \times 244$ PCA components to obtain an $8 \times 244$ weight matrix. We computed the magnitude for all the weights in the matrix, and then normalized them by the largest weight. We then computed the average weight for each of the neurons by taking the average for each column. Neurons were identified as non-linear mixed selective (NMS), linear mixed selective (LMS), and classically selective (CS) based on a two-way ANOVA with independent variables for target location and trial epoch.

**Cross-temporal decoding**. In order to assess the stability of the population code, we used data at each time point to train a decoder based on Linear discriminant analysis (LDA), built using the classify function in MATLAB, and tested the decoder on data from other time points. One minor difference in the current work was that instead of using normalized z-scored data to decode the target locations, we used projections in the 58 dimensional PCA space (the data we performed the optimization on) identified using data from Delay 1 and Delay 2 data (full space), projections in a 49-dimensional null space (for Fig. 4), or the projections in the optimized subspace. As we used projections to decode instead of normalized z-scored pseudo-population data, the performance of the decoder in this paper was higher than the results shown in Parthasarathy et al.[14]. For error trials, the decoder was trained on correct trials (similar to the other analyses, but only four out of the

seven target locations were used) and tested on error trials (in full space, subspace, and null space).

In order to compute the decoding performances in Fig. 2d, we averaged the cross-temporal performance for classifiers trained in the last 500 ms of Delay 1 (800–1300 ms after target onset) and tested on data from Delay 1 (labelled as $LP_{11}$) and Delay 2 (1800–2300 ms after target onset, labelled as $LP_{12}$). Similarly, we averaged the cross-temporal performance for classifiers trained in Delay 2 and tested on data from Delay 2 ($LP_{22}$) and Delay 1 ($LP_{21}$). This then allowed us to quantify the change in performance when decoding across delay periods. By using different subsets of the data for training and testing, we obtained distributions of performance accuracies that we were able to use for testing statistical significance (described below).

**State space analysis**. After projecting the responses in $D_1$ and $D_2$ into the 58-component PCA space, we computed the center of each of the seven clusters representing target locations in $D_1$ and $D_2$. We then computed the inter-delay distance between the centers of corresponding target locations, and used that to generate the values for the full space plotted in Fig. 2e. We also computed the distance between corresponding target locations in $D_1$ and during the presentation of the distractor. As a control, we divided both delay periods into an early 250 ms and a late 250 ms, and computed an equivalent intra-delay distance. In order to account for the variability within each cluster, we computed the average intra-cluster distances for all 14 clusters over 1000 bootstrapped samples. The intra-cluster distances were used to normalize both the inter-delay and intra-delay distances. By using different subsets of the data to form $D_1$ and $D_2$, we obtained distributions of both inter-delay and intra-delay distances that we were able to use for testing statistical significance (described below).

After transforming $D_1$ and $D_2$ into the subspace using the $U$ matrix, we repeated the steps described above to generate the values for the subspace plotted in Fig. 2e.

**Trajectory directions**. After projecting the responses in $D_1$ and $D_2$ into the 58-component PCA space, we computed the seven vectors that connected target locations in $D_1$ with the corresponding locations in $D_2$. We then used a measure known as phase locking value[28] (PLV) to quantify the similarity between the vectors. Briefly, this measure averaged the seven vectors together and computed the magnitude of the average vector. If the vectors were very similar, the PLV will be close to 1. If the vectors were quite dissimilar, the magnitude of the average vector will be close to 0. For comparison, we shuffled the target locations in $D_2$, and re-computed the seven vectors before computing the PLV. By using different subsets of data to form $D_1$ and $D_2$, we obtained distributions of PLVs that we were able to use for testing statistical significance (described below).

**Null space**. The $U$ matrix described above transformed the $D_1$ and $D_2$ vectors in the full space to the $SP1$ and $SP2$ vectors in the subspace. We then identified a null space where a basis set of vectors, $v$, returned $U(v) = 0$. This meant that the null space consisted of responses in the full space that were not captured in the subspace. For this, we used the *null* function in MATLAB to compute the null space given the $8 \times 58$ $U$ matrix. This returned a $50 \times 58$ $V$ matrix that defined the null space.

**Statistics**. We considered two bootstrapped distributions to be significantly different if the 95th percentile range of the two distributions did not overlap. We also computed an estimated *p*-value for this comparison using the following formula[29],

$$\frac{1 + X}{N + 1} \quad (2)$$

where X represents the number of overlapping data points between the two distributions and N represents the number of bootstraps. With this computation, and the $N = 1000$ bootstraps we used throughout the paper, two distributions with no overlap will result in a *p*-value <0.001, and two distributions with x% of overlap will result in a *p*-value ~x/100.

In addition to the estimated *p*-value, we also computed the effect size of the comparison using a measure known as Hedges' *g*, computed using the following formula[30].

$$\left( 1 - \frac{3}{4(n_1 + n_2) - 9} \right) \times \left( \frac{\overline{x_1} - \overline{x_2}}{s'} \right) \quad (3)$$

where

$$s' = \sqrt{\frac{(n_1 - 1)s_1^2 + (n_2 - 1)s_2^2}{n_1 + n_2 - 2}} \quad (4)$$

refers to the mean of each distribution, *n* refers to the length of each distribution, and *s* refers to the standard deviation of each distribution.

No statistical methods were used to pre-determine sample sizes, but our sample sizes are similar to those reported in previous publications. The majority of our analyzes made use of non-parametric permutation tests, and as such, did not make assumptions regarding the distribution of the data. No randomization was used during

the data collection, except in the selection of the target and distractor locations for each trial. Randomization was used extensively in the data analyzed to test for statistical significance. Data collection and analysis were not performed blind to the conditions of the experiments. No animals or data points were excluded from any of the analyses. Please see additional information in the Life Sciences Reporting Summary.

**Model.** In order to replicate the features we found in our neural data, we tested two types of artificial neural network models that were typically used to model working memory with persistent activity: (1) linear subspace model and (2) bump attractor model. We found that only the bump attractor model was able to replicate all the significant features found in our data. For the bump attractor model, we used $N = 80$ firing-rate units as the whole population, and used simplified discrete time equations to describe the dynamics of the population:

$$r_{t+1} = \varphi(W_{rec}r_t + W_{in}I_t + \sigma) \qquad (5)$$

where $r_t$ was the population firing rate at time $t$; $W_{rec}$ was the recurrent connection weight between units; $I_t$ was the external input at time $t$; $W_{in}$ was the loading weight of input signal to the population; $\sigma \sim N(0,0.1)$ was a noise term; and $\varphi(x)$ was a piecewise nonlinear activation function adopted from Wimmer et al.[17]:

$$\varphi(x) = \begin{cases} 0, & x < 0 \\ x^2, & 0 < x < 1 \\ \sqrt{4x - 3}, & x > 1 \end{cases} \qquad (6)$$

The matrix, $W_{rec}$, had a diagonal shape with stronger positive values near the diagonal, and weaker negative values elsewhere, such that only a few neighboring units were connected via excitatory weights to each other while being connected via inhibitory weights to the rest. In this way, a structured input signal to adjacent units was able to generate a local self-sustaining bump of activity. There were eight input units, representing the eight spatial target locations in the animal's task. For each input unit, the loading weight matrix, $W_{in}$, specified 10 adjacent units in the population to receive the input signal, and the loading population for each input unit were non-overlapping (Fig. 5a, different colors of stimuli). We also randomly chose $n$ individual units from the whole population as 'non-memory' units (Fig. 5a, green circles) to simulate adding information to the population that was different from the memory of the target location.

We tested a range of distractor activity levels relative to the target, and found that higher distractor activity led to higher distractor decoding accuracy, as expected. However, we reported in Parthasarathy et al.[14] that the distractor decoding performance was 1/3 of the target decoding performance. The red bar in Supplementary Fig. 9a indicated a range of distractor activity levels that replicated the lower distractor decoding performance compared to the target decoding performance. Within this range, we chose 0.2 as the distractor activity level. However, with that level of activity for the distractor on its own without any non-memory input, we did not observe any code-morphing. We believe this was because the weak distractor inputs alone caused only small shifts of the population activity in state space, which did not result in the population activity crossing the boundaries of the LDA cross-temporal decoding classifier trained with the target information in Delay 1 alone. This resulted in no reduction in the cross-temporal decoding performance in Delay 2, which was different from what we saw in our data. When we added the non-memory inputs, in each simulated trial, the same $n$ 'non-memory' units received inputs during the distractor period with strength, $s$, regardless of the distractor position. We tested different pairs of combinations of $n$ and $s$, and found that the pairs that successfully replicated code-morphing exhibited an anti-correlation within a range of $n$ and $s$, as depicted in the red circles in Supplementary Fig. 9b. We chose to use $n = 10$ and $s = 5$, which fell in the middle of the range of the red circles. We believe that the addition of the much stronger non-memory input resulted in the population activity crossing the boundaries of the classifier, resulting in the poor decoder performance we observed in Delay 2.

For the linear subspace model, we also used $N = 80$ units as the whole population, and the dynamics of the activity could be described as:

$$r_{t+1} = W_{rec}r_t + W_{in}I_t + \sigma \qquad (7)$$

where $r_t$ was the population firing rate at time $t$; $I_t$ was the external input at time $t$; $W_{in}$ was the loading weight of the input signal to the population; and $\sigma \sim N(0,0.1)$ was a noise term. We constructed the recurrent weight matrix $W_{rec}$ from eigendecomposition:

$$W_{rec} = Q\Lambda Q^{-1} \qquad (8)$$

where $Q$ was a random square matrix whose columns were the eigenvectors of $W_{rec}$, and $\Lambda$ was a diagonal matrix whose diagonal elements are the corresponding eigenvalues for each eigenvector. We specified the first nine eigenvalues in $\Lambda$ to be 1 (thus there were nine stable eigenvectors), and chose the rest of the eigenvalues randomly between 0 and 1 using a uniform distribution. For the input weight matrix $W_{in}$, we assigned eight stable eigenvectors to the eight target inputs, and one stable eigenvector to the non-memory input. The distractor inputs had the same input activity as did the target inputs, but with a lower magnitude (0.2 compared to target). As all the input activity corresponded to stable eigenvectors, all target information, distractor information, and non-memory information were maintained stably across time.

Similar to the bump attractor model, adding a weak distractor input did not result in code morphing in the Murray model (Supplementary Fig. 8a, b), while adding a strong non-memory input produced code morphing (Supplementary Fig. 8c). However, the code morphing was driven primarily by neurons with LMS instead of neurons with NMS. We believe this was because of two factors. First, we added a single non-memory input, which resulted in an identical translation in state space for all target locations. This meant that for each single neuron, the change from Delay 1 to Delay 2 would be the same for all target locations, and thus the neuron will be classified as a neuron with LMS (illustrated in Supplementary Fig. 8d). Second, the prevalence of neurons with NMS in the bump attractor model was because the firing rate of the neurons were limited by a saturating non-linear transfer function, while the Murray model, being a linear model with no restrictions on the firing rate of the neurons, ended up with a solution using neurons with LMS. If instead, we added two or more different non-memory inputs, or if we used a linear transfer function in the bump attractor model, we believe the Murray model and the bump attractor will produce the same results.

**Reporting summary.** Further information on research design is available in the Nature Research Reporting Summary linked to this article.

## Data availability
The data that support the findings of this study are available from the corresponding authors upon reasonable request.

## Code availability
A code package for performing the optimization is available at https://github.com/aishu1803/Subspace.git

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

## Acknowledgements

We thank Apoorva Bhandari and Andrew Tan for discussions and suggestions on this work. This work was supported by startup grants from the Ministry of Education Tier 1 Academic Research Fund and SINAPSE to C.L., a grant from the NUS-NUHS Memory Networks Program to S.-C.Y., a grant from the Ministry of Education Tier 2 Academic Research Fund to C.L. and S-C.Y. (MOE2016-T2-2-117), and a grant from the Ministry of Education Tier 3 Academic Research Fund to C.L. and S-C.Y. (MOE2017-T3-1-002).

## Author contributions

S-C.Y., A.P., and C.L., conceptualized the analysis framework. A.P., C.T., and R.H. performed the analysis. L-F.C. provided advice on the optimization technique. S.-C.Y. and C.L. guided the data analysis. All authors discussed the results, and A.P., S-C.Y., and C.L. wrote thepaper.

## Competing interests

The authors declare no competing interests.
