## [Peer Review File · Nature Communications]

Editorial Note: This manuscript has been previously reviewed at another journal that is not operating a transparent peer review scheme. This document only contains reviewer comments and rebuttal letters for versions considered at Nature Communications .

Reviewers' Comments:

Reviewer #1:

Remarks to the Author:

In this article, Parthasarathy and colleagues investigate the representation of stimulus information in the activity of prefrontal cortical neurons, during working memory. This group has previously described that stimulus information represented in neural activity morphs after the presentation of a behaviorally-irrelevant stimulus. They now show that there exists a low-dimensional subspace in which working memory remains stable before, throughout, and after such a stimulus presentation. They also consider the implantation of such a code, and find that a modified bump-attractor network best reproduces the experimental data.

I previously reviewed this article for another journal, and I found it to be an interesting and important contribution to the field. The revised manuscript has addressed all of my concerns. I also considered the responses of the authors to the other reviewers. I found the revisions to be responsive to all issues raised and ultimately, convincing. Most importantly, the reviewers cast doubt on the objective function that the authors used to obtain the subspace minimizing the distance between the two delay period representations. The authors have adopted the reviewer's recommendation, which practically required them to redo the entire analysis and figures. The authors have also revised the text to address the second major reviewer criticism about the alternative models they considered and whether alternative mechanisms can be ruled out. They have qualified their text and conclusions, appropriately.

I have only a few minor comments on the manuscript:

1. I would recommend that Figure R1, which was included in the response to the reviewers to show existence of the stable subspace in both monkeys, be added to the supplementary material. Two reviewers raised this issue. The figure is convincing and helpful.
2. I found the revised text to be a bit terse. Some of the revisions appear to have broken the flow of the narrative. I would suggest the authors use subheadings in the results and discussion section, which are allowed by the journal.
3. In the previous review, a referee raised a reasonable question on whether a bump attractor network with dedicated units to represent distractor information is plausible. I consider it a strength of the model that a single, interconnected network can represent both types of information. The authors may wish to point out that current state-of-the-art models separate inputs from different sources to be represented by different units, and clusters of network units being activated during different tasks emerge even if not specified explicitly: Yang et al. Nature Neuroscience, 22:297-306, 2019.

Reviewer #2:

Remarks to the Author:

I appreciate that the authors have improved the analysis of the stable coding space in the revised manuscript (thanks to a change in objective function suggested by one of the reviewers). My

comments have been addressed adequately, either through improving the representation of the results or through removing non-essential parts of the manuscript that I thought were problematic (e.g. RNNs in the modeling part).

Two of my main issues remain.

(1) Regarding novelty and impact: as I had pointed out previously, the only new result here is the co-existence of code-morphing and stable subspace, essentially clarifying the results of their previous paper (ref 14).

(2) I am skeptical about the new concept of bump attractor models with "non-memory" inputs that leave "spurious bumps". I agree with the authors that their model can capture some of features of the data (in terms of population decoding). However, I still think that the idea of non-memory inputs seems like a tweak to the model (in order to be able to account for a specific feature of the experimental data) rather than a conceptual advance.

In sum, while I definitely think this work should be disseminated these two points limit my enthusiasm for recommending it for a high-profile journal like Nat Commun.

Minor issue: The analysis that that was included in response to my previous comment that a mnemonic decoder would also learn a stable subspace throughout the trial was unclear to me. Was the mnemonic decoder trained with data from both delay intervals? Or separately for delay 1 and 2? Why exactly doesn't it find the stable subspace?

Reviewer #1 (Remarks to the Author):

In this article, Parthasarathy and colleagues investigate the representation of stimulus information in the activity of prefrontal cortical neurons, during working memory. This group has previously described that stimulus information represented in neural activity morphs after the presentation of a behaviorally-irrelevant stimulus. They now show that there exists a low-dimensional subspace in which working memory remains stable before, throughout, and after such a stimulus presentation. They also consider the implantation of such a code, and find that a modified bump-attractor network best reproduces the experimental data.

I previously reviewed this article for another journal, and I found it to be an interesting and important contribution to the field. The revised manuscript has addressed all of my concerns. I also considered the responses of the authors to the other reviewers. I found the revisions to be responsive to all issues raised and ultimately, convincing. Most importantly, the reviewers cast doubt on the objective function that the authors used to obtain the subspace minimizing the distance between the two delay period representations. The authors have adopted the reviewer's recommendation, which practically required them to redo the entire analysis and figures. The authors have also revised the text to address the second major reviewer criticism about the alternative models they considered and whether alternative mechanisms can be ruled out. They have qualified their text and conclusions, appropriately.

I have only a few minor comments on the manuscript:

1. I would recommend that Figure R1, which was included in the response to the reviewers to show existence of the stable subspace in both monkeys, be added to the supplementary material. Two reviewers raised this issue. The figure is convincing and helpful.

We thank the reviewer for this suggestion. We have added the result as Supplementary Figure 1.

2. I found the revised text to be a bit terse. Some of the revisions appear to have broken the flow of the narrative. I would suggest the authors use subheadings in the results and discussion section, which are allowed by the journal.

We thank the reviewer for pointing this out. We have added suitable subheadings to the manuscript.

3. In the previous review, a referee raised a reasonable question on whether a bump attractor network with dedicated units to represent distractor information is plausible. I consider it a strength of the model that a single, interconnected network can represent both types of information. The authors may wish to point out that current state-of-the-art models separate inputs from different sources to be represented by different units, and clusters of network units being activated during different tasks emerge even if not specified explicitly: Yang et al. Nature Neuroscience, 22:297-306, 2019.

We would like to thank the reviewer for this positive comment. We have highlighted this unique property of our model to represent both target and distractor information within the same population as opposed to the Yang et al. (2019) model in Lines 320 – 322.

Reviewer #2 (Remarks to the Author):

I appreciate that the authors have improved the analysis of the stable coding space in the revised manuscript (thanks to a change in objective function suggested by one of the reviewers). My comments have been addressed adequately, either through improving the representation of the results or through removing non-essential parts of the manuscript that I thought were problematic (e.g. RNNs in the modeling part).

Two of my main issues remain.

(1) Regarding novelty and impact: as I had pointed out previously, the only new result here is the co-existence of code-morphing and stable subspace, essentially clarifying the results of their previous paper (ref 14).

Yes, we agree that the main novelty in this paper is the co-existence of code-morphing and a stable subspace, which we found to be an elegant solution to how information can be read out over time in a population exhibiting code morphing.

(2) I am skeptical about the new concept of bump attractor models with “non-memory” inputs that leave "spurious bumps". I agree with the authors that their model can capture some of features of the data (in terms of population decoding). However, I still think that the idea of non-memory inputs seems like a tweak to the model (in order to be able to account for a specific feature of the experimental data) rather than a conceptual advance.

We would like to clarify that the “non-memory” input is not spurious in the sense that it is not related to the task. We admit we are not clear what this “non-memory” input is related to (although we offer some suggestions in the text), but we believe it to be task-related as it takes the same form in each trial.

In sum, while I definitely think this work should be disseminated these two points limit my enthusiasm for recommending it for a high-profile journal like Nat Commun.

Minor issue: The analysis that that was included in response to my previous comment that a mnemonic decoder would also learn a stable subspace throughout the trial was unclear to me. Was the mnemonic decoder trained with data from both delay intervals? Or separately for delay 1 and 2? Why exactly doesn't it find the stable subspace?

The task used in Murray *et al.* (2017) was limited to one visual stimulus, and the delay period following this stimulus was used to compute the mnemonic subspace. We followed the same method, and used the data recorded in the epochs following target presentation until the end of the trial (800 ms after target onset until 2500 ms after target onset) to compute the mnemonic subspace. However, we believe the presence of distractor, and therefore the presence of the morphed population code in Delay 2, prevented that method from identifying a stable subspace. Specifically, the mnemonic subspace (Supplementary Figure 6) in this case did not exhibit overlapping Delay 1 and Delay 2 clusters that was seen in the subspace computed through optimization (Fig 2).